# Smartphone Technology to Remotely Measure Postural Sway during Double- and Single-Leg Squats in Adults with Femoroacetabular Impingement and Those with No Hip Pain

**DOI:** 10.3390/s23115101

**Published:** 2023-05-26

**Authors:** Charlotte J. Marshall, Charlotte Ganderton, Adam Feltham, Doa El-Ansary, Adrian Pranata, John O’Donnell, Amir Takla, Phong Tran, Nilmini Wickramasinghe, Oren Tirosh

**Affiliations:** 1School of Health Sciences, Swinburne University of Technology, Hawthorn 3122, Australia; 2School of Clinical Medicine, Shanghai University of Medicine and Health Sciences, Shanghai 201318, China; 3Department of Surgery, School of Medicine, University of Melbourne, Parkville 3052, Australia; 4School of Health and Biomedical Sciences, RMIT University, Bundoora 3083, Australia; 5Hip Arthroscopy Australia, 21 Erin Street, Richmond 3121, Australia; 6Department of Orthopaedic Surgery, Western Health, Footscray Hospital, Footscray 3011, Australia

**Keywords:** femoroacetabular impingement syndrome, smartphone telehealth, postural sway, squat assessment

## Abstract

Background: The COVID-19 pandemic has accelerated the demand for utilising telehealth as a major mode of healthcare delivery, with increasing interest in the use of tele-platforms for remote patient assessment. In this context, the use of smartphone technology to measure squat performance in people with and without femoroacetabular impingement (FAI) syndrome has not been reported yet. We developed a novel smartphone application, the TelePhysio app, which allows the clinician to remotely connect to the patient’s device and measure their squat performance in real time using the smartphone inertial sensors. The aim of this study was to investigate the association and test–retest reliability of the TelePhysio app in measuring postural sway performance during a double-leg (DLS) and single-leg (SLS) squat task. In addition, the study investigated the ability of TelePhysio to detect differences in DLS and SLS performance between people with FAI and without hip pain. Methods: A total of 30 healthy (nfemales = 12) young adults and 10 adults (nfemales = 2) with diagnosed FAI syndrome participated in the study. Healthy participants performed DLS and SLS on force plates in our laboratory, and remotely in their homes using the TelePhysio smartphone application. Sway measurements were compared using the centre of pressure (CoP) and smartphone inertial sensor data. A total of 10 participants with FAI (nfemales = 2) performed the squat assessments remotely. Four sway measurements in each axis (x, y, and z) were computed from the TelePhysio inertial sensors: (1) average acceleration magnitude from the mean (aam), (2) root-mean-square acceleration (rms), (3) range acceleration (r), and (4) approximate entropy (apen), with lower values indicating that the movement is more regular, repetitive, and predictable. Differences in TelePhysio squat sway data were compared between DLS and SLS, and between healthy and FAI adults, using analysis of variance with significance set at 0.05. Results: The TelePhysio aam measurements on the x- and y-axes had significant large correlations with the CoP measurements (r = 0.56 and r = 0.71, respectively). The TelePhysio aam measurements demonstrated moderate to substantial between-session reliability values of 0.73 (95% CI 0.62–0.81), 0.85 (95% CI 0.79–0.91), and 0.73 (95% CI 0.62–0.82) for aamx, aamy, and aamz, respectively. The DLS of the FAI participants showed significantly lower aam and apen values in the medio-lateral direction compared to the healthy DLS, healthy SLS, and FAI SLS groups (aam = 0.13, 0.19, 0.29, and 0.29, respectively; and apen = 0.33, 0.45, 0.52, and 0.48, respectively). In the anterior–posterior direction, healthy DLS showed significantly greater aam values compared to the healthy SLS, FAI DLS, and FAI SLS groups (1.26, 0.61, 0.68, and 0.35, respectively). Conclusions: The TelePhysio app is a valid and reliable method of measuring postural control during DLS and SLS tasks. The application is capable of distinguishing performance levels between DLS and SLS tasks, and between healthy and FAI young adults. The DLS task is sufficient to distinguish the level of performance between healthy and FAI adults. This study validates the use of smartphone technology as a tele-assessment clinical tool for remote squat assessment.

## 1. Introduction

A squat is a functional and dynamic movement that is used in clinical and sport-specific settings to assess leg extension strength [1], range of motion [1], and movement quality [2]. The double-leg squat (DLS) is performed when both feet are hip-width and on the ground while the single-leg squat (SLS) is performed when standing on one foot pointed straight forward and the other, non-weight-bearing leg raised in the air above ground. Healthcare professionals use squats in the clinical assessment of common hip musculoskeletal conditions, as it is estimated that 10% to 20% of athletic or recreational childhood injuries are hip-related, and 6% of adult sports injuries occur at the hip [3,4]. Furthermore, lower extremity musculoskeletal conditions, including femoroacetabular impingement (FAI) syndrome and hip arthroscopy, have been shown to affect the quality of the squat movement [5].

Balance performance during squats is one of the outcome measures used to identify lower extremity impairment [6,7,8,9]. The anterior–posterior (AP) and medio-lateral (ML) displacement and velocity of the centre of pressure (CoP) measured with a force plate have been used as outcome measures to identify balance impairment during squats. For example, greater CoP displacement and velocities in both AP and ML directions have been reported in hip chondropathy patients compared to healthy controls [8]. Similarly, single-leg squat performance in people with hypertonic-saline-injection-induced hip pain, showed a significant decrease in ML range of 13% (*p* = 0.03) and AP range of 21% (*p* = 0.01), and a decrease in AP velocity of 13% (*p* = 0.03) when compared with no pain measures. Unfortunately, force plates are often unavailable in clinical settings because they are expensive, cumbersome, and require a specific measurement setup. Moreover, the exponential increase in digitalised health services during the COVID-19 pandemic [10] further compounded the difficulties of using a force plate as a tele-assessment tool for measuring squat performance.

To address the paucity of remote tele-assessment solutions that quantify squat performance, we designed the TelePhysio network platform—a web-based repository system coupled with the motion sensor data captured from an inertial measuring unit (IMU) in a smartphone. TelePhysio allows the clinician to connect remotely to the patient’s smartphone sensors from their personal web browser and collect live patient motion data while instructing the patient to perform the specific functional task. The TelePhysio platform has already proven effective in measuring the hip range of motion [11]. The use of a portable smartphone to measure standing balance is not new [12,13], but the capability to measure dynamic squat performance remotely through tele-assessment, to our knowledge, has not been investigated previously.

Including TelePhysio in clinical care programs for patients presents an opportunity to significantly impact accessibility and equity, reducing the need for clinicians and patients to seek specialised healthcare by travelling to metropolitan areas for their ongoing management. COVID-19 has highlighted the need for a workforce with a strong digital health capability that delivers higher quality care and positive consumer experiences to engage patients as empowered participants in their own care. Telehealth platforms, such as TelePhysio, with validated and meaningful measures of outcomes inclusive of the squat test, may enable care outside of hospital settings, in the community or at home, improve operational efficiencies, and enhance clinical workflows and the automation of repetitive tasks, thus improving the ability to monitor, diagnose and manage patients. Research has shown that patient perspectives on telemedicine during the COVID-19 pandemic are positive and favourable due to factors such as convenience, lack of travel, scheduling ease, and time saved [14]. With improvements in communication technology, such as the development of 5G, patient rehabilitation and functional performance outside the hospitals and clinics becomes more viable, enabling people with physical and functional disorders to access specialists [15]. This system may also improve the flow and sharing of information throughout the health system. However, care must be taken with personal information data as network transfer is vulnerable to cyber-attacks [16].

The aims of this study were (1) to evaluate the association of TelePhysio sway measurements with force plate CoP measures of postural sway during the DLS and SLS tasks, (2) to examine the between-sessions reliability of TelePhysio sway measurements for the evaluation of DLS and SLS performance, and (3) to explore the use of TelePhysio for detecting differences in DLS and SLS sway between healthy and FAI adults.

## 2. Materials and Methods

### 2.1. Participants

A total of 30 healthy adults (n_females_ = 12) and 10 FAI participants (n_females_ = 2) were recruited for the study. All participants were English-speaking and aged 19–53 years. The inclusion criteria for participants were as follows: no history of hip pain/injury or surgery, no history of lower limb injuries over the last three months, and a negative Flexion-Adduction-Internal Rotation (FADIR) test. The FAI participants in this study had hip pain with combined hip flexion, adduction, and internal rotation (FADIR test); a range of motion restriction and pain with hip internal rotation at 90 degrees of hip flexion; and had diagnostic imaging (X-ray) that indicated a CAM lesion (alpha angle ≥ 60°). Ethical approval was granted by the Swinburne University Human Ethics Committee (ref: 20215539-8106). All participants provided signed written informed consent prior to testing.

### 2.2. Instrumentation

Double- and single-leg squats were measured using the smartphone TelePhysio app. TelePhysio is our original tele-assessment platform that uses smartphone motion sensors to allow clinicians to measure and assess the real-time functional performance of their patients in a remote and objective way (Figure 1). At the beginning of the assessment, the examiner remotely connects using their web browser to the participant’s TelePhysio app, installed on the participant’s smartphone that is inserted in a pouch strapped around their waist, positioned at their lower back (Figure 1). All participants used their personal smartphones; thus, the smartphones were not of any specific brand and included both iPhone (*n* = 25; models 8, X, SE, 11, 12, and 13) and Android (*n* = 15; models Samsung Galaxy, Huawei P10, and Google pixel) models. At the end of the assessment, both the examiner and the participant were able to log into the web-based application and review the outcomes. A force plate (600 × 400 × 100 mm, 1000 Hz, Kistler Group, Switzerland) was used as the validation gold standard instrument to assess the association between the TelePhysio sensor output and force plate CoP output. A validation trial was performed at our biomechanics laboratory only for the healthy participants during the first testing session.

### 2.3. Protocol

Healthy participants were assessed on two occasions, with an average of 6 days (max 12 days) between sessions. The first session was conducted face-to-face in the biomechanics laboratory and the second session was conducted remotely at the participant’s home. The duration of the laboratory session was 45 min, and the at-home session was 30 min long. In both sessions, the participant turned on the TelePhysio app and strapped the smartphone around their waist positioned at the lower back. The examiner then connected remotely to the TelePhysio app from their personal web browser and collected motion data whilst the participant performed the DBL and SLS tasks. Once the specific task was completed, the examiner uploaded the smartphone sensor’s data to the web-based cloud application for later processing and analysis. During the assessment, voice communication between the examiner and participant was maintained via Zoom (Zoom Video Communications, Inc., San Jose, CA, USA). During the first testing session in the biomechanics laboratory, the participant performed the squat standing on the force plate sampled at 1000 Hz to measure the AP and ML CoP displacement. For FAI participants, the testing session was performed remotely at the participant’s home.

Both DLS and SLS were performed barefoot at self-selected speeds to be more representative of movement evaluations in the clinical setting. During the DLS, participants raised their arms to shoulder height, with their fingertips pointing forward and palms facing the floor (see Figure 1B1); for the SLS, participants were asked to keep their arms across their chests (see Figure 1C1). They were then instructed to “squat as low as possible while keeping your feet/foot firmly in contact with the floor at all times”. Participants performed 3 squats to maximal depth in each repetition. A successful trial was a squat where the participant’s feet remained in contact with the ground throughout the movement. There were 2 min breaks between each squat task.

### 2.4. Data Analysis

All accelerometer and gyroscope signals from the smartphone were processed using R-Statistics [17]. Outcome measurements from TelePhysio included: (1) average acceleration magnitude from the mean (aam), (2) root-mean-square acceleration (rms), (3) range acceleration (r) as the distance between the maximum and minimum accelerometry signal, and (4) approximate entropy (apen), with lower values indicating that the movement is more regular, repetitive, and predictable. All measurements were calculated for the x-, y-, and z-axes (sagittal, transverse, and frontal axes, respectively) as illustrated in Figure 1. The acceleration signal processing and calculation of the aam, rms, and r measurements were performed as described previously [18]. Briefly, the gravity component was eliminated by subtracting the acceleration signal mean, followed by a zero-phase Butterworth high-pass filter at 0.3 Hz, and then a third-order Savitzky–Golay smoothing filter with frames of 41 points [18]. Approximate entropy was calculated to quantify the amount of regularity and unpredictability of fluctuations, as described earlier for dynamic tasks such as walking [19].

The x-axis angular velocity from the smartphone gyroscope signal was used to identify the beginning and end of each squat repetition. Similar to Beyea et al.’s [20] study, the angular velocity was processed by first filtering with a Butterworth low-pass filter (10 Hz, 4th order, and zero-lag), followed by rectifying the signal and normalizing to its maximum peak value. Figure 2 shows the identification of a sample 3-repetition double-leg squat performance. The first angular velocity peak represents descending and the second peak ascending; thus, two adjunct peaks represent one squat repetition.

The CoP data from the force plate were processed and calculated as described previously [21]. Signals were low-pass filtered at 20 Hz, using a 4th order Butterworth filter, to calculate the CoP range (mm), root mean square (mm), and velocity (mm s^−1^) in the AP and ML directions.

### 2.5. Statistical Analysis

The association between CoP measurements derived from the force plate and the measurements derived from the smartphone TelePhysio app was assessed with Spearman rank-order correlations for all DLS and SLS conditions. Correlation coefficients of 0.1 were considered small, 0.3 were considered moderate, and 0.5 were considered large [22]. To explore between-sessions reliability, the intraclass correlation coefficient (ICC) was used. The interpretation of each ICC (2,1) adhered to the original definitions by Landis and Koch [23], as follows: 0.00 to 0.20, slight correlation; 0.21 to 0.40, fair correlation; 0.41 to 0.60, moderate correlation; 0.61 to 0.80, substantial correlation; and 0.81 to 1.00, almost perfect correlation. The SEM was calculated for each measurement modality as an additional measure of absolute reliability. This was calculated as described by Atkinson and Nevill [24], as $SEM=SD\sqrt{1-ICC}$, where SD is the standard deviation. Lower SEM values indicate better absolute reliability. In addition, the minimal detectable change (MDC) was calculated to investigate the measurement error, in order to show the range within which the amount of change in the two measured values obtained by repeated measurements is due to measurement error. Changes greater than the MDC are judged to be “true changes”. The MDC was calculated using the following formula: $MDC=1.96\sqrt{2SEM}$. For all analyses, the means of the measurements were compared using a one-way analysis of variance (ANOVA) with a Benjamini and Hochberg adjusted correction [25] as a post hoc analysis, if there was statistical significance at a 0.05 level. The Benjamini and Hochberg adjustment controls the false discovery rate—the expected proportion of false discoveries amongst the rejected hypotheses. The false discovery rate is a less stringent condition than the family-wise error rate, so these methods are more powerful than the others. All analyses were performed using the free software Statistical Package R version 4.2.1 (https://www.r-project.org/ (accessed on July 2022)) with the significance level set at 0.05.

## 3. Results

The 30 healthy participants’ mean age, weight, and height were 29.00 ± 4.84 years (22–39 years), 73.39 ± 12.32 kg (45–95 kg), and 1.71 ± 0.09 m (1.49–1.91 m), respectively. Only one healthy participant reported their left leg as the dominant kicking leg. The 10 FAI participants’ mean age, weight, and height were 37.30 ± 10.82 years (19–53 years), 80.40 ± 11.54 kg (59–98 kg), and 1.77 ± 0.08 m, respectively (1.65–1.88 m). Only one FAI participant reported their left leg as the dominant leg, and eight FAI participants had their right hip as the injured side.

### 3.1. Association of TelePhysio Sway Measurements with Force Plate CoP Measurements

The first aim of the study was to measure the association between the TelePhysio sway measures and the force plate CoP measurements. Table 1 shows the Spearman correlation results for the DLS condition. The CoP mean and max velocity measurements were shown to have the best correlations with TelePhysio. In the ML sway, the TelePhysio aamx measurement had significant large correlations with the CoP measurements of range, RMS, mean velocity, and max velocity (0.56, −0.53, 0.48, and 0.41, respectively). In the AP sway, the CoP mean velocity had large correlations with the TelePhysio measurements on the y-axis of aamy, rmsy, and ry (0.71, 0.68, and 0.62, respectively), and on the z-axis of rmsz, rz, and apenz (0.49, 0.53, and 0.44, respectively).

Table 2 shows the Spearman correlation results for the SLS condition. The CoP mean and max velocity measurements were shown to have the best correlations (medium to large) with the TelePhysio measurements. In the ML sway, the CoP mean velocity had significant correlations with the TelePhysio x-axis measurements of 0.43, 0.46, and 0.37 for aamx, rmsx, and rx, respectively. Similarly, CoP max velocity in ML had significant correlations with the TelePhysio x-axis measurements of 0.50, 0.53, and 0.42 for aamx, rmsx, and rx, respectively. In the AP sway, the CoP mean velocity had large correlations with the TelePhysio y-axis measurements of aamy, rmsy, ry, and apeny (0.66, 0.64, 0.47, and 0.53, respectively). The AP CoP max velocity showed medium and large correlations with the smartphone x-axis measurements of 0.50, 0.54, and 0.52 for aamx, rmsx, and rx, respectively, and on the y-axis, of 0.43, 0.45, and 0.46 for aamy, rmsy, and ry, respectively.

### 3.2. Between-Sessions Reliability of TelePhysio Sway Measurements

Table 3 shows the between-sessions reliability of the TelePhysio sway measurements. The average acceleration magnitude from the mean showed the highest ICC values of 0.73 (95% CI 0.62–0.81), 0.85 (95% CI 0.79–0.91), and 0.73 (95% CI 0.62–0.82) for aamx, aamy, and aamz, respectively, which are considered as moderate to substantial correlation. Similarly, the approximate entropy measurements that indicate movement regularity showed moderate to substantial agreement values of 0.72 (95% CI 0.59–0.81), 0.68 (95% CI 0.51–0.79), and 0.77 (95% CI 0.67–0.84) for apenx, apeny, and apenz, respectively. The SEM for the AAM values ranged from 0.06 to 0.26 m/s^−2^ with an average MDC of 1.07 m/s^−2^ (0.66 m/s^−2^ to 1.42 m/s^−2^).

### 3.3. Differences in DLS and SLS Sway between Healthy and FAI Adults

The third specific aim of this study was to determine the sensitivity of several TelePhysio sway measurements to changes in squat stability by hip injury (healthy and FAI) and under two conditions (DLS and SLS). Figure 3, Figure 4 and Figure 5 show the comparison of sway measurements between the healthy DLS, healthy SLS, FAI DLS, and FAI-injured leg SLS conditions on the x-, y-, and z-axes, respectively. Overall people with FAI showed significantly more regular and less ML sway during the DLS task, while in the AP direction, healthy controls showed significantly less regular and more sway during the DLS task. Figure 3 shows the TelePhysio sway measurements on the x-axis for the healthy DLS, healthy SLS, FAI DLS, and FAI-injured leg SLS conditions. In all the TelePhysio measurements on the x-axis, the FAI group showed significantly lower values in the DLS condition compared to the injured SLS, healthy DLS, and healthy SLS groups, indicating less and more regular sway. The DLS condition in healthy controls showed significantly lower aamx and rmsx values (0.19 ± 0.05 and 1.00 ± 0.004, respectively) compared to healthy SLS (0.29 ± 0.12 and 1.02 ± 0.09, respectively) and injured SLS (0.29 ± 0.13 and 1.02 ± 0.02, respectively).

Figure 4 shows the TelePhysio sway measurements on the y-axis for the healthy DLS, healthy SLS, FAI DLS, and FAI-injured leg SLS conditions. The DLS condition in healthy controls showed significantly greater aamy, rmsy, and ry values (0.99 ± 0.39, 1.24 ± 0.24, and 5.73 ± 2.59, respectively) compared to healthy SLS (0.56 ± 0.31, 1.15 ± 0.55, and 4.10 ± 2.47, respectively), FAI DLS (0.46 ± 0.21, 1.04 ± 0.04, and 2.65 ± 1.08, respectively), and injured SLS (0.43 ± 0.12, 1.04 ± 0.02, and 3.05 ± 0.88, respectively).

Figure 5 shows the TelePhysio sway measurements on the z-axis for the healthy DLS, healthy SLS, FAI DLS, and FAI-injured leg SLS conditions. The DLS condition in healthy controls showed significantly greater aamz, rmsz, and rz values (1.26 ± 0.59, 1.35 ± 0.29, and 5.75 ± 1.81, respectively) compared to healthy SLS (0.61 ± 0.39, 1.01 ± 0.09, and 2.93 ± 1.23, respectively), FAI DLS (0.68 ± 0.39, 1.12 ± 0.16, and 3.36 ± 1.48, respectively), and injured SLS (0.35 ± 0.14, 1.02 ± 0.02, and 1.94 ± 0.74, respectively). The FAI-injured SLS condition showed significantly lower aamz, rmsz, and rz values compared to all other conditions.

## 4. Discussion

The main aim of this study was to investigate the validity and between-sessions reliability of a real-time remote tele-assessment application (TelePhysio) that measures postural sway during double- and single-leg squat tasks as representative of commonly performed clinical functional tests to identify lower extremity impairment. We also evaluated the ability of TelePhysio to distinguish changes in sway measurements related to the squat condition (double- and single-leg) and hip injury (healthy vs. FAI). The results of this study may help guide healthcare practitioners to collect objective digitised data remotely and support the transition from face-to-face to telehealth services (i.e., due to the COVID-19 pandemic).

The results from this study support our hypothesis that a smartphone provides valid measurements of sway during the DLS and SLS conditions. However, the association was not perfect. The TelePhysio sway measurements were found to have a significantly large association with the CoP measurements in both the DLS and SLS conditions. In ML sway during the DLS condition, the TelePhysio average acceleration magnitude (aamx) and root-mean-square (rmsx) sway measurements showed a significantly large agreement of 0.53 and 0.51 with the CoP rms values, respectively. Similarly, in AP sway, the CoP mean velocity measurement showed a significantly large agreement of 0.71 and 0.68 with the TelePhysio aamy and rmsy, respectively. Similar associations were found for the SLS condition, having a large agreement in ML sway of 0.5 and 0.53 for ammx and rmsx with the CoP velocity max measurement, and in AP sway, 0.66 and 0.64 for aamy and rmsy with the CoP mean velocity measure. These results align with past studies that found that rms derived from smartphones is most comparable with gold standard devices for measuring postural stability [26,27,28]. The average acceleration magnitude was also comparable to the CoP measurements, which suggests a high association between the two measuring devices when using this measurement. The imperfect association between the smartphone acceleration and CoP measures, however, is not surprising since the measurements are measuring different aspects of stability [29]. The smartphone measures sway at the approximate centre of mass (COM) while the CoP measurements are derived from the ground reactive forces that may be more reflective of the corrective actions taken by the individual. Winter [30] proposed that acceleration measurement at the COM may be better as it is an approximate measurement of body sway about the COM. Another explanation for the imperfect association between the two sway measurement methods (smartphone and force plate) may be that during the squat movement, the individual does not sway strictly as an inverted pendulum due to the trunk tilt, and thus movement measured by TelePhysio at the pelvic level may not directly reflect the pressure movement between the feet and the force plate. If the body was moving like an inverted pendulum, a correlation close to 1 would be expected between trunk acceleration and the COP displacement [30].

The between-sessions reliability of the TelePhysio measurements in this study was found to be substantial to almost perfect. The average acceleration magnitude was found to have the greatest ICC values of 0.73, 0.85, and 0.73 for aamx, aamy, and aamz, respectively. The approximate entropy measurement that indicates movement regularity also showed substantial reliability values of 0.72, 0.68, and 0.77 for apenx, apeny, and apenz, respectively. To our knowledge, this is the first study reporting the reliability of postural sway measurements during squats using smartphone technology.

As stated above, one of the aims of this study was to explore the ability of the smartphone to identify the level of performance during DLS and SLS tasks and to differentiate between people with and without hip pain. In this study, people with FAI were chosen for comparison with the performance of healthy controls. The FAI group were chosen because the squat task is a task that helps clinicians evaluate movement impairment in the FAI syndrome [31]. Other groups with hip injuries could be considered in a future study. The average acceleration magnitude (aam) and the approximate entropy (apen) measurements that showed substantial between-sessions reliability also showed the ability to significantly differentiate the level of squat performance between DLS healthy, SLS healthy, DLS FAI, and injured leg SLS FAI. Other measurements such as the rms and the range (r) also showed the ability to significantly differentiate between conditions. For ML sway, the FAI group showed significantly lower aamx and apenx values during the DLS condition compared to the healthy DLS, healthy SLS, and FAI-injured SLS conditions. These results show that the DLS task is performed by people with FAI with minimum ML sway, and the movement is well controlled and highly regular. Both the healthy SLS and FIA SLS conditions showed significantly greater aamx and apenx values when compared to the DLS conditions (healthy and FAI), but they were not significantly different from each other. Our results are different from Malloy et al.’s [31] findings showing that SLS but not DLS tasks differentiate between movement patterns in people with FAI syndrome and those without hip pain. Malloy et al. [31] reported that during the SLS task, adults with FAI syndrome squatted more slowly and with less peak hip adduction and greater hip movements when compared to healthy controls. The differences in the conclusions between the studies may relate to differences in the measurement methods. Our study examined the magnitude and regulation (complexity) of postural sway, while Malloy et al. [31] examined the kinematics and kinetics of hip joint movement. It is possible that despite differences in hip kinematics between FAI and healthy control adults during the SLS task, as reported by Malloy et al. [31], both groups controlled their ML sway with similar consistency and strategy, as reported by this study; otherwise, small errors with minor fluctuations that are less regulated will result in balance loss and falls.

Our results showing similar values of ML sway in SLS performance between healthy and FAI participants may be surprising as we expected that the FAI group will find it more challenging to perform SLS with the injured leg, and would therefore experience greater ML sway during SLS compared to healthy controls. Our results, however, did show that in the AP direction, the FAI group had significantly lower aamz and apenz values during SLS compared to healthy controls, which indicates less and more regulated AP sway with the injured leg. This finding is similar to the findings of other studies that explored movement patterns in participants with joint pain. For example, in a field study with butchers [32], neck–shoulder discomfort was associated with less motor variability and greater movement regularity. Similarly, in studies with unilateral knee injuries, the injured leg exhibited less motor variability and greater regularity during gait than the non-injured knee of the same individual [33,34]. These observations lend support to the hypothesis that decreased movement variability and increased regularity is a result of the pain constraining the movements within tighter boundaries so that pain can be reduced [35], potentially similar to the strategy that our FAI used when squatting with their injured hip.

Our study has several limitations. First, participants used a range of personal smartphone devices, and this may introduce potential errors due to data sampling from variations in smartphone devices, operating software platforms, and motion sensors. Grouios et al. [36] compared acceleration data from three smartphones, including iPhone 12 Pro Max, Samsung Galaxy S21 Ultra, and Huawei P Smart, and concluded that the mean acceleration data were not statistically different between the devices. This result was similar to other research studies [37,38] that compared accelerometer sensor performance between commercially available smartphones, suggesting that modern smartphone accelerometers can be used for measuring human movement and be employed in clinical research. Second, the study population was limited to healthy and FAI young adults, and future research needs to widen the range of participants with respect to age and lower extremity injury. Third, we asked the participants to squat as low as possible and we did not standardise the depth of the squat, which resulted in different squat depths between participants. This may affect the outcome measures, although the study showed that the smartphone system can recognise and distinguish between squat type and level of impairment. The fourth limitation to the proposed method may be the need for the patient to have a smartphone and be connected to the internet. According to the Australian Bureau of Statistics, in 2018, 86% of households in Australia had the potential to connect to the Internet and 91% had smartphones [39]. Nevertheless, the smartphone system and the protocol were found to be valid, reliable, and discriminating, which should encourage its use in clinical settings.

## 5. Conclusions

Remote tele-assessment using a smartphone application to measure postural sway during double-leg and single-leg squats has moderate to substantial between-session reliability and can significantly differentiate between levels of performance in adults that are asymptomatic and those presenting with FAI. Specifically for individuals with FAI syndrome, the DLS task is sufficient to differentiate levels of performance, and thus the SLS task may not be necessary when experiencing difficulties during SLS with the injured leg. Therefore, we recommend using the DLS task for assessing people with hip pain. The use of a smartphone to remotely assess squat performance is an appropriate and effective alternative to a face-to-face assessment. The use of a smartphone to remotely assess postural sway during squats has the potential to improve patient outcomes by advancing equitable access to optimal management. The method and technology used in this study represent a paradigm shift, allowing clinicians to remotely perform and quantify the much-needed squat performance assessment of their patients, potentially reaching more patients at a reduced cost by reducing patient travel. TelePhysio offers a new mode of care that can be integrated into current telehealth consultation methods for people with FAI. The TelePhysio squat assessment solution fits within the routine clinical care of people with FAI and additionally allows assessments to be performed remotely, enabling greater access to more patients, including those in regional and remote areas.

## Figures and Tables

**Figure 1 sensors-23-05101-f001:**
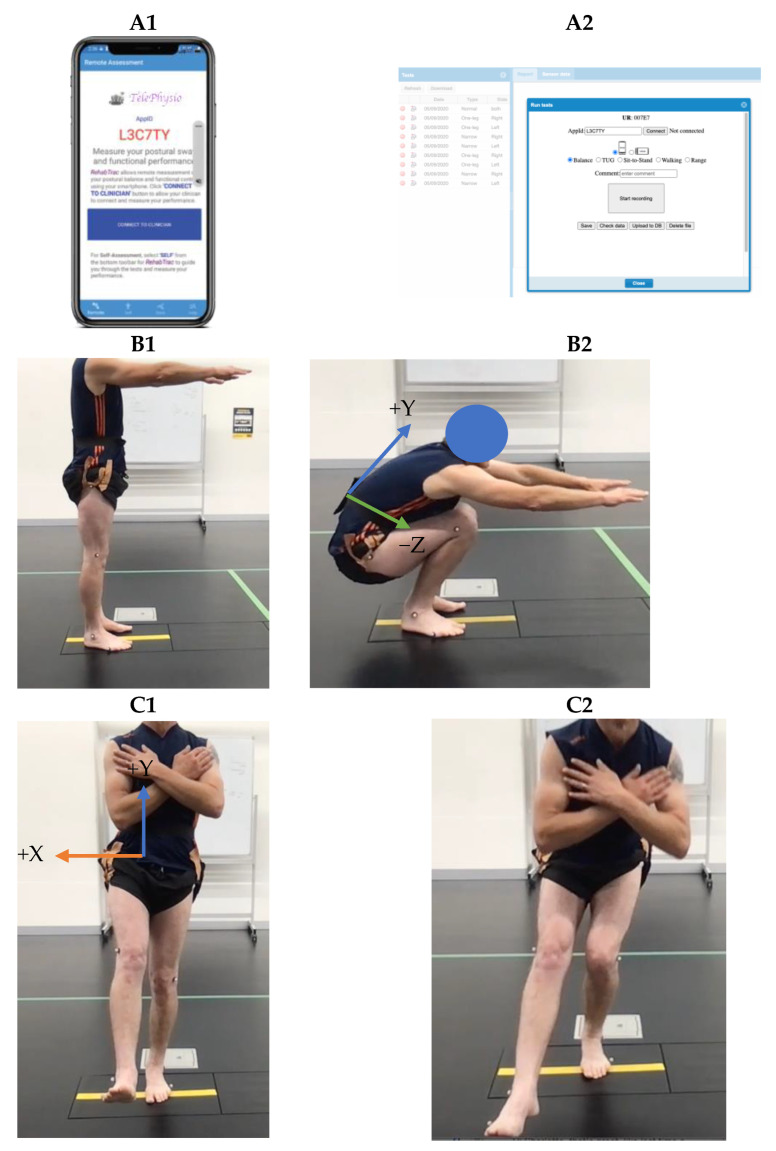
The TelePhysio app that is installed on the participant’s smartphone (**A1**) and the web-based interface (**A2**) that is controlled from the clinician’s web browser. Double-leg (**B1**,**B2**) and single-leg (**C1**,**C2**) squat tasks. Smartphone inside the black pouch strapped around the waist at the lower back. Arrows indicate the smartphone accelerometer axes.

**Figure 2 sensors-23-05101-f002:**
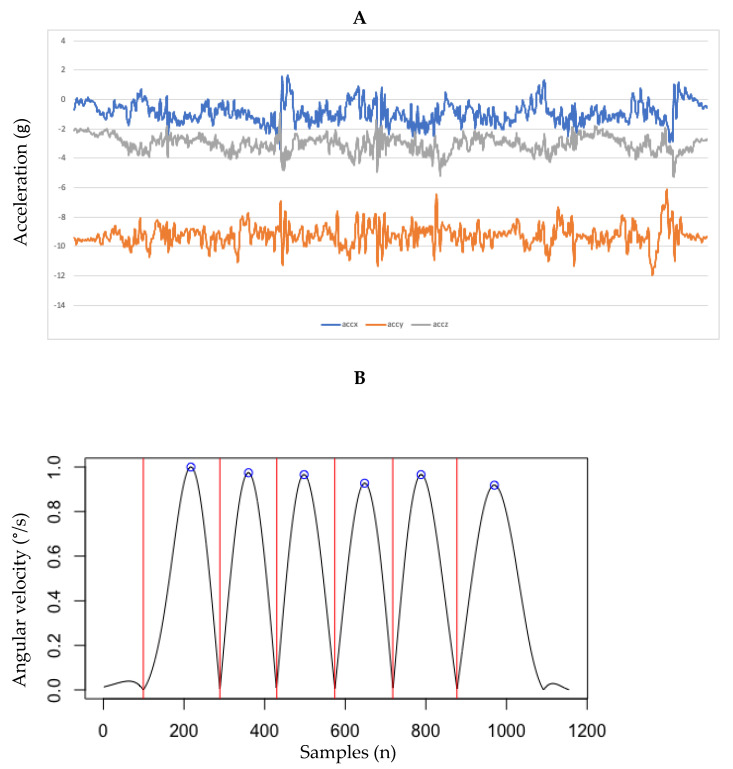
(**A**) Raw acceleration in x, y, and z direction (accx, accy, and accz, respectively, and (**B**) identification of squat repetition using angular velocity. Angular velocity was processed by first filtering with a Butterworth lowpass filter (10 Hz, 4th order, and zerolag), followed by rectifying the signal and normalizing it to its maximum peak value. The first red vertical line represents the beginning of the squat by lowering the body. The second red vertical line represents raising the body. The last vertical line represents the end of the 3 squat repetitions.

**Figure 3 sensors-23-05101-f003:**
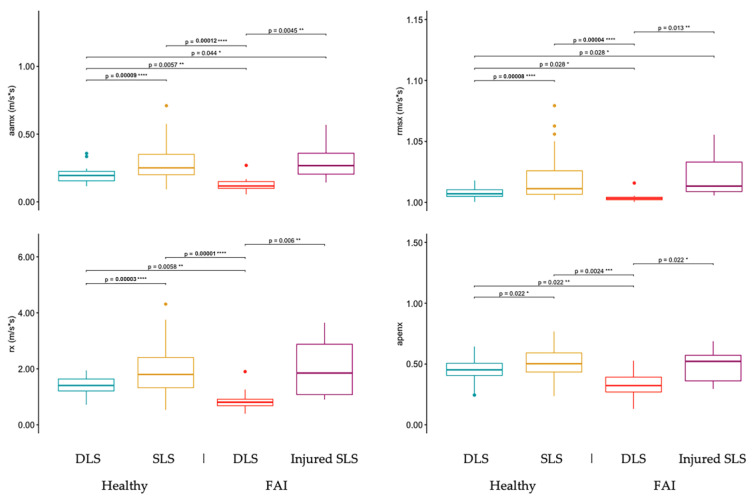
Smartphone sway measurements on the sagittal axis (x, medio-lateral) for double-leg squats (DLS) and single-leg squats (SLS) in both healthy and FAI adult participants. The SLS for FAI participants is with the injured leg. Sway measurements include average acceleration magnitude from the mean (aamx), root-mean-square acceleration (rmsx), range acceleration (rx), and approximate entropy (apenx). *, **, ***, and **** denote *p*-values of *p* < 0.05, *p* < 0.01, *p* < 0.005, and *p* < 0.001, respectively.

**Figure 4 sensors-23-05101-f004:**
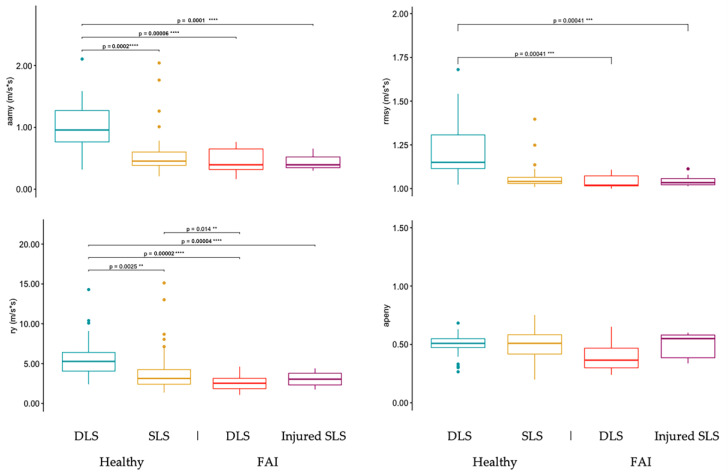
Smartphone sway measurements on the transverse axis (y) for double-leg squats (DLS) and single-leg squats (SLS) in both healthy and FAI adult participants. The SLS for FAI participants is with the injured leg. Sway measurements include average acceleration magnitude from the mean (aamy), root-mean-square acceleration (rmsy), range acceleration (ry), and approximate entropy (apeny). **, ***, and **** denote *p*-values of *p* < 0.05, *p* < 0.01, *p* < 0.005, and *p* < 0.001, respectively.

**Figure 5 sensors-23-05101-f005:**
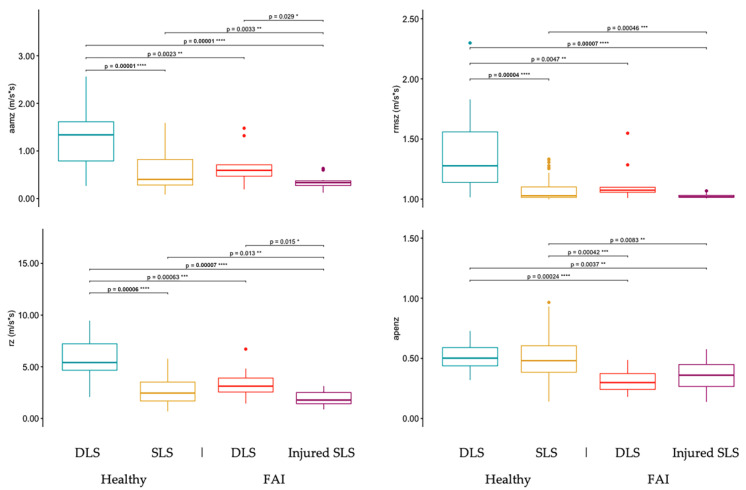
Smartphone sway measurements on the frontal axis (z, anterior–posterior) for double-leg squats (DLS) and single-leg squats (SLS) in both healthy and FAI adult participants. The SLS for FAI participants is with the injured leg. Sway measurements include average acceleration magnitude from the mean (aamz), root-mean-square acceleration (rmsz), range acceleration (rz), and approximate entropy (apenz). *, **, ***, and **** denote *p*-values of *p* < 0.05, *p* < 0.01, *p* < 0.005, and *p* < 0.001, respectively.

**Table 1 sensors-23-05101-t001:** Correlations between the smartphone TelePhysio app and force plate CoP measurements of postural sway during double-leg squats. Large correlations (<0.5) are in bold.

		CoP Medio-Lateral				Cop Anterior–Posterior			
		Range	RMS	Velocity Mean	Velocity Max	Range	RMS	Velocity Mean	Velocity Max
**Smartphone Sway Measurement**	aamx	**0.56** **	**−0.53** **	0.48 **	0.41 **	0.27	−0.07	0.47 **	0.33
	rmsx	0.37	**−0.51** **	0.16	0.20	0.01	−0.03	0.43 *	0.06
	rx	0.31 *	−0.12	0.08	0.23	0.26	−0.03	0.20	−0.02
	apenx	0.31	−0.24	0.29	0.44 **	0.00	−0.16	0.42 *	0.19
	aamy	0.04	−0.01	0.19	0.28	0.19	−0.04	**0.71** **	0.36
	rmsy	0.06	−0.03	0.15	0.31 *	0.15	−0.01	**0.68** **	0.32
	ry	0.06	−0.04	0.32	0.35	0.23	−0.01	**0.62** **	0.29
	apeny	0.05	0.03	0.27	0.34 *	0.07	−0.09	0.28	0.23
	aamz	0.06	−0.17	0.16	0.21	−0.11	−0.27	0.36	0.25
	rmsz	0.02	−0.16	0.25	0.21	0.03	−0.14	0.49 **	0.32
	rz	0.02	−0.12	0.31	0.30	0.01	−0.12	**0.53** **	0.36
	apenz	0.11	−0.06	0.33	0.47 **	−0.02	−0.06	0.44 *	0.20

* *p*-values were significant (*p* < 0.05); ** *p*-values were significant (*p* < 0.01).

**Table 2 sensors-23-05101-t002:** Correlations between the smartphone TelePhysio app and the force plate CoP measurements of postural sway during the single-leg squat. Large correlations (<0.5) are in bold.

		CoP Medio-Lateral				Cop Anterior–Posterior			
		Range	RMS	Velocity Mean	Velocity Max	Range	RMS	Velocity Mean	Velocity Max
**Smartphone Sway Measurement**	aamx	0.15	−0.03	0.43 **	**0.50** **	014	0.05	0.40 **	**0.50** **
	rmsx	0.23	0.02	0.46 **	**0.53** **	0.11	0.05	0.45 **	**0.54** **
	rx	0.17	0.03	0.37 **	0.42 **	0.18	0.16	0.32 **	**0.52** **
	apenx	0.14	−0.02	0.23	0.27	0.21	−0.03	0.41 **	0.45 **
	aamy	0.17	−0.11	0.43 **	0.34	0.34 *	−0.10	**0.66** **	0.43 **
	rmsy	0.14	−0.13	0.38 **	0.38	0.36 *	−0.14	**0.64** **	0.45 **
	ry	0.15	−0.06	0.35 *	0.34	0.36 *	−0.11	0.47 **	0.46 **
	apeny	0.15	−0.04	0.10	0.29	0.26 *	0.01	**0.53** **	0.42 **
	aamz	−0.08	−0.06	0.05	0.08	0.03	0.12	0.17	0.23 *
	rmsz	−0.09	−0.07	0.03	−0.09	0.04	0.08	0.12	0.33 *
	rz	−0.11	−0.11	0.11	−0.04	0.07	0.16	0.22	0.31 *
	apenz	0.26	−0.05	0.24	0.23	0.22	−0.16	0.35 *	0.37 **

* *p*-values were significant (*p* < 0.05); ** *p*-values were significant (*p* < 0.01).

**Table 3 sensors-23-05101-t003:** Comparison of smartphone (TelePhysio) measurements for healthy participants in the laboratory and at home: means ± standard deviations, one-way ANOVA *p*-value, ICC and 95% confidence intervals, SEM, and MDC. Significant (*p* < 0.05) are in bold.

		Laboratory Mean ± SD	Home Mean ± SD	*p*-Values	ICC −95%–+95% CI	SEM	MDC
**Smartphone Sway Measurement**	aamx	0.26 ± 0.11	0.26 ± 0.11	0.76	0.73 0.62–0.81	0.06	0.66
	aamy	0.71 ± 0.40	0.77 ± 0.46	**0.02**	0.85 0.76–0.91	0.17	1.13
	aamz	0.83 ± 0.56	0.76 ± 0.45	0.08	0.73 0.62–0.82	0.26	1.42
	rmsx	1.02 ± 0.02	1.02 ± 0.01	0.96	0.63 0.48–0.74	0.01	0.26
	rmsy	1.19 ± 0.47	1.22 ± 0.57	0.52	0.45 0.27–0.60	0.39	1.72
	rmsz	1.17 ± 0.23	1.13 ± 0.14	**0.01**	0.65 0.50–0.75	0.11	0.93
	rx	1.82 ± 0.76	1.92 ± 0.88	0.23	0.48 0.31–0.62	0.59	2.13
	ry	4.65 ± 2.62	4.96 ± 3.07	0.10	0.80 0.71–0.87	1.27	3.12
	rz	3.88 ± 1.96	3.63 ± 1.81	0.05	0.79 0.70–0.86	0.86	2.57
	apenx	0.49 ± 0.13	0.52 ± 0.10	**0.002**	0.72 0.59–0.81	0.06	0.68
	apeny	0.49 ± 0.11	0.53 ± 0.10	**0.002**	0.68 0.51–0.79	0.06	0.68
	apenz	0.52 ± 0.15	0.50 ± 0.13	0.09	0.77 0.67–0.84	0.07	0.72

## Data Availability

Data available on request.

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
