# Peer review of "Smartphone Technology to Remotely Measure Postural Sway during Double- and Single-Leg Squats in Adults with Femoroacetabular Impingement and Those with No Hip Pain"

_sensors, 2023, doi:10.3390/s23115101_

Round 1

Reviewer 1 Report

This work have present a smartphone application (TelePhysio app) that measure squat performance using inertial sensors of smartphone. To study the ability of the TelePhysio, the paper investigated the association and test reliability to the app to measure postural sway performance during a double-leg and single-leg squat task. Although, the paper provide contributions, there are some issues needed to be answered and improved before publication

Some specific comments:

Some abbreviations did not be defined before using, such as ML . Please carefully check.

The authors should clearly state contributions of this work in the introduction.

To better understand, the authors should show some samples of accelerometers and gyroscopes used for data analysis.

Please show some screenshots of the TelePhysio app.

What is the difference between inertial sensors in iPhone and Android phones? Please discuss.

Axis titles in Figure 2 is not completed. Please check.

What is limitation of this proposed method.

Author Response

  • Some abbreviations did not be defined before using, such as ML. Please carefully check.

Author response: Appreciate your comment. We re-checked and updated abbreviations per comment.

  • The authors should clearly state contributions of this work in the introduction.

Author response: As advised we included a significant and impact statement at the end of the introduction, see lines 89-100. In addition, in the conclusions we provided additional sentences to conclude the impact and paradigm shift of the proposed method and technology (see lines 442-449).

  • To better understand, the authors should show some samples of accelerometers and gyroscopes used for data analysis.

Author response: It was not clear what the reviewer is asking regarding “To better understand” i.e what to understand, and “some samples” i.e is it sample images of the accelerometer sensors? OR sample figures of the raw accelerometer signal data? As images of the sensors are impossible as they are embedded inside the smartphone, we therefore added and included in figure 2 the accelerometers raw signal data (A) with the already presented gyroscope signal (B) in figure 2.

  • Please show some screenshots of the TelePhysio app.

Author response: we added and included TelePhysio screenshots in figure 1 (A1 and A2).

  • What is the difference between inertial sensors in iPhone and Android phones? Please discuss.

Author response: we included few sentences in the limitations paragraph regarding the literature findings on differences in acceleration data between smartphone manufactures. See lines 414-419.

  • Axis titles in Figure 2 is not completed. Please check.

Author response: As suggested we included angular velocity units and sample (n) in axis titles

  • What is limitation of this proposed method.

Author response: We updated the limitations section, see lines 411-430.

Reviewer 2 Report

The paper discusses a smartphone app remotely measuring their squat performance in real time.

The paper is well organized and described.  A minor revision is required.

Strengths: measurement approach and measurement protocol.  

Points of weakness: description of the state of the art, discussion of sensor technology

Actions to do:

According to the weaknesses, I suggest to improve the paper by answering to these points:

·       The authors should describe deeply the inertial sensor technology by comparing/discussing similar technology presented in literature and by highlighting differences;   

·       Please provide a photo zooming the adopted sensors by describing it.

·       Conclusion section should be improved (I suggest to cut something from the abstract which is too long to add in the conclusion section);

·       Please summarize the results of Fig. 3, 4 and 5;

·       Further references should be added in the introduction section about possible perspective of wearable health sensors, big data and artificial intelligence applications (perspectives in telemedicine), such as:

https://doi.org/10.3390/s23031678

DOI: 10.1109/MetroInd4.0IoT48571.2020.9138258

https://doi.org/10.3390/s23052519

DOI: 10.23919/AEIT.2018.8577362

https://doi.org/10.3390/app13031394

https://doi.org/10.3390/diagnostics12122964

Minor remarks:

Please read again the whole paper to correcting some typos errors.  

Good

Author Response

  • The authors should describe deeply the inertial sensor technology by comparing/discussing similar technology presented in literature and by highlighting differences;  

Author response: we included additional information in the limitations highlighting the differences in acceleration data between smartphone technologies found in the literature. See lines 411- 430. We would like to note that in this paper our focus is on the use of smartphone to measure squat task, thus, we did not provide extensive review about inertial sensor technology that can be further found in previous publications.

  • Please provide a photo zooming the adopted sensors by describing it.

Author response: we are a bit confused regarding this request of “a photo zooming the adopted sensors”. The study used data collected from smartphone sensors that are embedded in the smartphone, thus it is impossible for us to provide a photo of the adopted sensors. The list of the smartphone used in this study are mentioned in “instrumentation” sub-heading, see lines 128-130.

  • Conclusion section should be improved (I suggest to cut something from the abstract which is too long to add in the conclusion section);

Author response: we found it difficult to understand what the reviewer means by “improve” and to understand the reviewer request of what exactly they are interested to include in the conclusion. However, we included additional information (see lines 442-449) to conclude the impact and benefit of the proposed method and technology.

  • Please summarize the results of Fig. 3, 4 and 5;

Author response: as suggested we included a brief summary of the figure, see lines 279-283.

  • Further references should be added in the introduction section about possible perspective of wearable health sensors, big data and artificial intelligence applications (perspectives in telemedicine), such as:

https://doi.org/10.3390/s23031678

DOI: 10.1109/MetroInd4.0IoT48571.2020.9138258

https://doi.org/10.3390/s23052519

DOI: 10.23919/AEIT.2018.8577362

https://doi.org/10.3390/app13031394

https://doi.org/10.3390/diagnostics12122964

Author response: as suggested we included the references in the introduction in a paragraph mentioning the impact, patient perspective, 5G technology, and security, see lines 85-102.

Minor remarks:

  • Please read again the whole paper to correcting some typos errors.  

Author response: appreciate the reviewer comment. The manuscript was again proofed read for typo errors. Note that some may be related to differences between US, UK, and Australian words e.g digitise vs digitize.

Round 2

Reviewer 1 Report

The authors have addressed all the previous comments.